# A Systematic Investigation of Lipid Transfer Proteins Involved in Male Fertility and Other Biological Processes in Maize

**DOI:** 10.3390/ijms24021660

**Published:** 2023-01-14

**Authors:** Chaowei Fang, Suowei Wu, Ziwen Li, Shuangshuang Pan, Yuru Wu, Xueli An, Yan Long, Xun Wei, Xiangyuan Wan

**Affiliations:** Zhongzhi International Institute of Agricultural Biosciences, Research Center of Biology and Agriculture, School of Chemistry and Biological Engineering, University of Science and Technology Beijing (USTB), Beijing 100096, China

**Keywords:** lipid transfer proteins, anther and pollen development, vegetive organ development, seed development, stress response, substrate analysis, maize

## Abstract

Plant lipid transfer proteins (LTPs) play essential roles in various biological processes, including anther and pollen development, vegetative organ development, seed development and germination, and stress response, but the research progress varies greatly among *Arabidopsis*, rice and maize. Here, we presented a preliminary introduction and characterization of the whole 65 LTP genes in maize, and performed a phylogenetic tree and gene ontology analysis of the LTP family members in maize. We compared the research progresses of the reported LTP genes involved in male fertility and other biological processes in *Arabidopsis* and rice, and thus provided some implications for their maize orthologs, which will provide useful clues for the investigation of LTP transporters in maize. We predicted the functions of LTP genes based on bioinformatic analyses of their spatiotemporal expression patterns by using RNA-seq and qRT-PCR assays. Finally, we discussed the advances and challenges in substrate identification of plant LTPs, and presented the future research directions of LTPs in plants. This study provides a basic framework for functional research and the potential application of LTPs in multiple plants, especially for male sterility research and application in maize.

## 1. Introduction

Lipid transfer proteins (LTPs), also named non-specific LTPs (nsLTPs), defined by their ability to facilitate transfer of non-specific phospholipids and fatty acids between membranes in vitro, widely exist in all land plants but not in algae and other organisms. Therefore, LTPs are considered as key proteins for plant survival on and colonization of land [1,2]. LTPs are small, basic proteins characterized by eight conserved cysteine residues, low molecular weight (usually between 6.5 to 10 kDa), and a high content of four or five a-helices [3]. The LTPs are stabilized by four conserved disulfide bridges formed by an eight-Cys motif (8CM) with the general form C–Xn–C–Xn–CC–Xn–CXC–Xn–C–Xn–C, resulting in a central hydrophobic cavity suitable for the binding of various lipids [4]. According to the sequence similarity, the LTPs can be classified into 10 types, including five major types (LTP1, LTP2, LTPc, LTPd, and LTPg) and five minor types (LTPe, LTPf, LTPh, LTPj, and LTPk) [4]. Most LTP proteins include an N-terminal signal peptide and an LTP domain, and some also possess a glycophosphatidylinositol (GPI) motif (exit in LTPg), and the signal peptide and GPI motif are required for LTP subcellular localization and biological function [4,5]. Recent studies have shown that many intracellular LTPs localize on different organelles, forming a shuttle, bridge or tube that links donor and acceptor compartments to transport specific substrates [6,7]. LTPs were characterized as mediators to the transport of various types of lipids between different organelles [8,9], and involved in the transport of lipids in the intra-, extra-, and inter-cellular spaces [10,11].

To date, plant LTPs have been reported to be involved in numerous biological processes, such as anther and pollen development, vegetative organ development, seed development and germination, and biotic and abiotic stress response [12]. A total of 63, 80, and 84 LTP members have been predicted in maize, rice, and *Arabidopsis*, respectively [4,13], and plenty of LTP genes have been functionally characterized in rice and *Arabidopsis* involved in a wide range of biological functions, including anther cuticle and pollen wall formation [5,14,15,16], seed development [17,18,19,20], cell expansion and plant growth [17,21,22,23,24,25], and biotic and abiotic stress response [26,27,28,29,30]. Until now, at least 36 LTP genes were reported in *Arabidopsis* and rice, and 12 of them were essential for pollen development and male fertility. LTP proteins can function in the tapetum, extracellular space or microspore. For example, type III LTPs are reported to function as exine precursor distributors and exine constituents in *Arabidopsis* anther tapetum [14]. OsC6 is related to sporopollenin precursor translocation, and OsC6 is synergistic with the mediator OsLTP47 for pollen wall development in tapetum and extracellular space [16,31]. Further, OsEPAD1/OsLTPL94 functions as a microspore membrane recruiting protein to determine the pollen exine patterning [5,15]. However, only seven LTP transporters were characterized and four of them (Zmms44, ZmLTPg11, ZmLTPx2, and MZm3-3) were reported to be required for anther development and male fertility in maize [32,33,34,35,36,37]. Considering the functional conservation among homologous *LTP* genes of *Arabidopsis*, rice, and maize, the functional reports of LTP genes in *Arabidopsis* and rice will give helpful clues for investigating their homologs in maize. Furthermore, although the in vivo substrates of LTP transporters remain largely unknown, serval binding experiments of LTPs have been reported, such as an isotope labelling experiment in vivo, a protein-lipid overlay assay, or indirect imprecise measurements by using the Gas Chromatography–Mass Spectrometer (GC-MS) system [21,33,38].

Here, we updated the number and nomenclature of the maize *LTP* gene family, and carried out the phylogenetic tree and gene ontology analysis of LTPs in maize, rice, and *Arabidopsis*. The homologous and bioinformatic analysis methods were used to predict the functions of maize *LTP* genes, especially in anther development and male fertility. Furthermore, we summarized the progress of the substrate identification of LTPs, and discussed the future research directions and potential applications of LTP proteins in plants.

## 2. Results

### 2.1. The Nomenclature and Characterization of Maize LTP Genes

The 63 maize *LTP* genes have been divided into six types, including type 1, type 2, type C, type D, type G, and a unique type X, while most of their functions remain obscure [13]. Based on the updated maize B73 V5 reference genome (Zm-B73-REFERENCE-NAM-5.0) in the MaizeGDB (www.maizegdb.org, accessed on 25 October 2022) website, we identified 65 *ZmLTP* genes, which is different from the previous report because of different versions of the maize reference genome. For example, the gene model of *ZmLTP1.2* is *GRMZM2G010868* and *Zm00001d044686* in B73 V3 (B73 RefGen_v3) and B73 V4 (Zm-B73-REFERENCE-GRAMENE-4.0), respectively, but it corresponds to two gene models (*Zm00001eb163900* and *Zm00001eb163910*) in B73 V5, and thus they were named *ZmLTP1.2-1* and *ZmLTP1.2-2*. All the 65 *ZmLTP* genes are distributed to the ten chromosomes of maize with variable numbers, from ten *ZmLTPs* on chromosome 1 to two on chromosome 9 (Figure 1). The in silico mapping information will facilitate the gene cloning and evolution study of the *ZmLTP* genes.

Furthermore, the RNA-seq data revealing spatial expression patterns of maize LTP genes are retrieved from MaizeGDB (Table 1). Notably, *ZmLTP1.5*, *ZmLTP2.9*, *ZmLTPc1* (*Zmms44*), *ZmLTPc2* (*Zmnthr3*), *ZmLTPd9*, and *ZmLTPx1* are specially expressed in tassel, the *ZmLTPg20* is specially expressed in anther, *ZmLTPx2* is specially expressed in endosperm, and *ZmLTPd7* and *ZmLTPg12* are specially expressed in roots (Table 1), which will provide useful clues to explore their biological functions in different organs’ development. Other characteristics of the 65 *ZmLTP* genes, including the corresponding gene models in the B73 reference genome (B73v3 and B73v4), genome physical locus, topology, theoretical pI using Expasy (https://web.expasy.org/protparam/, accessed on 25 October 2022), and subcellular localization prediction using Cell-PLoc 2.0 (http://www.csbio.sjtu.edu.cn/bioinf/Cell-PLoc-2/, accessed on 25 October 2022), are listed in Appendix A. In a word, the chromosome locations, spatial expression patterns, and other characteristics are useful to explore the function and evolution of maize LTPs in the future.

### 2.2. Phylogenetic Analysis and Classification of LTPs in Maize

To analyze the phylogenetic relationship of the LTP gene family, the 65, 80, and 84 LTP protein sequences in maize, rice, and *Arabidopsis* were used to construct the phylogenetic tree by using the maximum likelihood method and the MEGA7 program. All the identified LTPs can be divided into three groups (namely group 1 to 3) in the phylogenetic tree (Figure 2). Group 1 can be further classified into six clades (Clades 1-1 to 1-6), and contains 91 LTPs, including 28 in maize, 31 in rice, and 32 in *Arabidopsis*. Among them, *OsLTPg25* (previously named as *OsC6*) [16], *OsLTPg28* (previously named as *OsLTP47*) [31], *AtLTPg3*, and *AtLTPg4* [17] were reported to be involved in pollen or anther development. Group 2 contains 89 LTPs, including 25 in maize, 35 in rice, and 29 in *Arabidopsis*, which can be further classified into five clades (Clades 2-1 to 2-5). The largest number of LTPs related to male sterility were reported in group 2, such as *AtLTPc3*, *AtLTPc1* [14], and their orthologs *OsLTPc1* (previously named as *OsC4*) [39,40], *ZmLTPc1* (previously named as *Zmms44*) [32], and *ZmLTPc2* (previously named as *MZm3-3/athr3*) [34]. In addition, *OsLTPd11* (previously named as *OsDIL/OsLTP6*) [41,42], *AtLTP1.7* (previously named as *AtLTP12*) [43], and *AtLTP1.8* (previously named as *AtLTP5*) [44,45] were reported to be involved in anther development. Group 3 can be further classified into three clades (Clades 3-1 to 3-3), and contains 91 LTPs, including 28 in maize, 31 in rice, and 32 in *Arabidopsis*. *ZmLTPx2*, *ZmLTPg11* [33], and their orthologs *OsLTPg29* (also named as *OsLTPL94* or *OsEPAD1*) [5,15] were reported to participate in anther and pollen wall development. Interestingly, there are only 19 *Arabidopsis* LTP genes in the clade 3-2 (Figure 2), indicating that these genes appear after the divergency of dicotyledons and monocotyledons during plant evolution. In general, the phylogenetic analysis results provide not only the phylogenetic relationship of the LTP genes, but also give some helpful information for exploring the functions of unknown LTP genes in maize, rice, and *Arabidopsis*.

### 2.3. Gene Ontology Analysis of LTPs in Maize

The Gene Ontology (GO) database has the highest annotation ratio compared with other analysis databases, such as CDD (Conserved Domain Database), KOG (Ortholog Group), or NR (Non-redundant database) [46]. In order to comprehend the molecular functions and putative pathways involving maize LTPs, GO enrichment analysis of the functional significance was performed. The 65 ZmLTPs were defined in 25 significant GO terms (Appendix A). The analysis showed that all the ZmLTPs were separated into two main categories (biological processes and molecular functions), which included 20 and 5 significant GO terms, respectively (Figure 3A). For the enriched biological processes, the 20 GO terms can be grouped into two classes, including biotic and abiotic stress response (10 GO terms: 0009627, 0009814, 0006955, 0045087, 0002376, 0098542, 0043207, 0051707, 0009607, and 0009605), and lipid and other substance transport and localization (10 GO terms: 0006869, 0071702, 0044765, 0006810, 0010876, 0033036, 1902578, 0051234, 0051179, and 005174). Notably, about 11 ZmLTPs were shown to participate in lipid and other substance transport and localization, which is in concordance with the molecular role of LTP in transporting hydrophobic molecules in vitro, suggesting that ZmLTPs play an important role in membrane components translocation. Five ZmLTPs were exhibited to participate in biotic and abiotic stress response (Figure 3A and Appendix A). This highlights the putative association of ZmLTPs in stress tolerance behavior of maize. In case of molecular functions, the 12 ZmLTPs were shown to participate in “lipid binding” (GO: 0008289) and five ZmLTPs were shown to be involved in binding of fatty acid, monocarboxylic acid, carboxylic acid and organic acid (GO: 0005504, 0033293, 0031406 and 0043177). In a word, the GO analysis indicated that ZmLTPs may be involved in diverse biological processes, such as lipid binding, lipid transport and localization, and biotic and abiotic stress response.

### 2.4. Functions of LTP Transporters in Rice and Arabidopsis and Their Implications in Maize

Until now, at least 25 and 11 LTP genes have been characterized in Arabidopsis and rice, respectively, but only seven have been reported in maize (Table 2). Considering the functional conservation of orthologous genes among multiple plant species, the functions of rice and *Arabidopsis* LTPs will provide important clues for exploring that of maize orthologs. Increasing evidence indicates that plant LTP transporters play important roles in anther and pollen development, vegetative and female organ development, seed development and germination, and biotic and abiotic stress response during plant biological processes (Table 2, Figure 4).

#### 2.4.1. Anther and Pollen Development

Plant lipids, including fatty acids and their derivatives, work as the essential building blocks of anther cuticle and pollen wall formation, and genic male sterility (GMS) would arise if the lipid metabolism is disrupted during anther and pollen development in plants [47]. Anther cuticles and pollen walls are composed of complex substances, including cutin, wax, and sporopollenin [47]. The precursors of sporopollenin, cutin, and wax are synthesized in anther tapetum, and catalyzed by a series of lipid metabolism-related GMS gene-encoding enzymes, such as *ZmMs25/ZmFAR1* encoding a plastid-localized fatty acyl reductase [48,49], *ZmMs33* encoding a glycerol-3-phosphate acyltransferase [50,51,52,53], *ZmMs30* encoding a GDSL lipase [54], and *ZmPKSB* encoding a polyketide synthase [55]. Then, the lipid precursors are transported into the pollen surface and anther outer wall by ABCGs such as ZmABCG26 and ZmMs13/ZmABCG2a [48,56,57], LTPs such as Zmms44 [32], and other transporters [47]. Loss functions of the anther-specific or high-expressed LTP genes often lead to defective anther cuticle and pollen wall formation and thus male sterility in plants [58].

**Table 2 ijms-24-01660-t002:** Functional classifications of the reported *LTP* genes in *Arabidopsis*, rice, and their orthologs in maize.

No.	Gene name	Gene ID	Gene Name in This Paper	Maize Orthologs	Expression Organs	Biological Functions	References
I. Anther development and male fertility							
1	*AtLTP5*	*At3g51600*	*AtLTP1.8*	*ZmLTP1.8-1*	Root, Shoot, Leaf, Pistil	Vegetative and pollen tube growth	[44,45]
2	*AtLTP12*	*At3g51590*	*AtLTP1.7*	*ZmLTP1.8-1*	Pollen	Restore the fertility of proline-deficient microspores	[43]
3	*AtLTPc1*	*At5g07230*	*AtLTPc1*	*ZmLTPc1*	Anther	Transport exine precursors from tapetal ER to microspore surface	[14]
4	*AtLTPc3*	*At5g62080*	*AtLTPc3*	*ZmLTPc1*	Anther	Transport exine precursors from tapetal ER to microspore surface	[14]
5	*OsC4*	*Os08g0546300*	*OsLTPc1*	*ZmLTPc1*	Anther	Related to tapetal PCD of anther	[39,40]
6	*ZmMs44*	*Zm00001d052736*	*ZmLTPc1*	–	Anther	Works as signal peptide and facilitates the secretion of lipids from tapetal cells into the locule	[32]
7	*AtLTPG3*	*At1g18280*	*AtLTPg3*	*ZmLTPg15*	Anther	Pollen grain development	[17]
8	*AtLTPG4*	*At1g27950*	*AtLTPg4*	*ZmLTPg4/* *ZmTLPg14*	Anther	Pollen grain development	[17]
9	*OsC6*	*Os11g0582500*	*OsLTPg25*	*ZmLTPx1*	Anther	Male reproductive development	[16]
10	*OsLTP47*	*Os01g0607100*	*OsLTPg28*	*ZmLTPg26*	Anther	Pollen wall development	[31]
11	*OsDIL/* *OsLTP6*	*Os10g0148000*	*OsLTPd11*	*ZmLTPg15/* *ZmTLP16*	Leaf, Lemma, Palea, Anther, Pistil	Drought stress, tapetal and anther sacs development	[41,42]
12	*OsLTPL94/ OsEPAD1*	*Os03g0663900*	*OsLTPg29*	*ZmLTPg11/* *ZmLTPx2*	Anther	Pollen exine formation	[5,15]
13	*ZmLTPg11*	*Zm00001d003600*	*ZmLTPg11*	–	Anther	The orthologs of TaMs1, but *zmltpg11zmltpx2* mutant is normal in pollen development	[33]
14	*ZmLTPx2*	*Zm00001d025467*	*ZmLTPx2*	–	Anther		
15	*MZm3-3*	*Zm00001d021226*	*ZmLTPc2*	–	Anther	Contribute to pollen coat formation	[34]
II. Vegetative organ and seed development							
1	*AtLTP1*	*At2g38540*	*AtLTP1.5*	*ZmLTP1.8-1*	Stem, leaf, and root	Export of lipids to the plant surface	[21,25,59]
2	*AtLTP2*	*At2g38530*	*AtLTP1.4*	*ZmLTP1.8-1*	Epidermal Cells	Play major structural roles	[60]
3	*AtLTP7*	*At2g15050*	*AtLTP1.1*	*ZmLTP1.8-1*	Leaf	–	[45]
4	*AtLTP6*	*At3g08770*	*AtLTP1.6*	*ZmLTP1.1*	Leaf and root	–	[45]
5	*OsPSD1/* *OsPTD1*	*Os01g0822900*	*OsLTP1.2*	*ZmLTP1.1*	Stem, Leaf	Cell development, plant height, sensitivity to temperature conditions	[22,24]
6	*AtLTPG2*	*At3g43720*	*AtLTPg21*	*ZmLTPg2*	Stem	Alkane and wax transport	[23]
7	*AtLTPG5*	*At1g36150*	*AtLTPg5*	*ZmLTPg25*	Seed	Seed coat permeability	[17]
8	*AtLTPG6*	*At1g55260*	*AtLTPg6*	*ZmLTPg9*	Seed	Cuticle development and seed coat suberization	[17]
9	*AtLTPG15*	*At2g48130*	*AtLTPG15*	*ZmLTPg5*	Root, seed	Seed coat permeability	[61]
10	*AtLTPG23*	*At4g08670*	*AtLTPg23*	*ZmLTPg5*	-	Related to suberin biosynthesis	[3]
11	*AtLTPG26*	*At4g14815*	*AtLTPg26*	*ZmLTPg7*	-	Related to suberin biosynthesis	[3]
12	*OsLTPL36*	*Os03g0369100*	*OsLTPd13*	*ZmLTPd6*	Seed	Seed quality, seed development and germination	[20]
13	*ZmBETL-9*	*Zm00001d041822*	*ZmLTPd6*	*-*	Endosperm	Transcribed in outer surface of developing endosperm	[35]
14	*AtEND1*	*At1g32280*	*AtLTPd9*	*ZmLTPg15/ ZmTLPg16*	Root, Leaf, Stem, Flower, Seed.	Gametophytic tissues and developing endosperm	[62]
15	*AtLSR1*	*At1g62500*	*AtLSR1*	–	Leaf	Regulate leaf senescence	[63]
III. Biotic and abiotic stress response							
1	*AtDIR1*	*At5g48485*	*AtLTPd1*	*ZmLTPd5*	Leaf	Transmission of mobile signal(s) during systemic acquired resistance	[64]
2	*AtDIR1-like*	*At5g48490*	*AtLTPd2*	*ZmLTPd5*	Leaf	Transmission of mobile signal(s) during systemic acquired resistance	[65]
3	*AtLTP3*	*At5g59320*	*AtLTP1.12*	*ZmLTP1.8-1*	Leaf, Flower, Silique, Root	Contributes to disease susceptibility	[19,66,67]
4	*AtLTP4*	*At5g59310*	*AtLTP1.11*	*ZmLTP1.8-1*	Shoot apex, Leaf, Root	Reduced susceptibility to *Pseudomonas* and down-regulation of ABA biosynthesis genes in *ltp3/ltp4* mutant	[45,66]
5	*AtLTPG1*	*At1g03103*	*AtLTPg1*	*ZmLTPg7/* *ZmTLPg13*	Stem, Leaf	Alter cuticular lipid composition, increase plastoglobulus, enhance susceptibility to infection by *Fungal Pathogen*	[27]
6	*OsLTP5*	*Os11g0115400*	*OsLTP1.22*	*ZmLTP1.2-2/ ZmLTP1.6*	Stem, Flower	Response to ABA, salicylic acid, and 16-hydroxypalmitic acid	[26]
7	*Zm-LTP*	*Zm00001d044686*	*ZmLTP1.2*	–	–	Binds to calmodulin (CaM) in a Ca^2+^-independent manner	[36]
8	*OsLTPL159*	*Os10g0505500*	*OsLTP2.11*	*ZmLTP2.4*	Seeding, root, node, Leaf, Sheath, Spikelet	Involved in cold tolerance at early seedling stage in rice	[30]
9	*OsLTP110*	*Os11g0115100*	*OsLTP1.9*	*ZmLTP1.7*	–	Inhibit germination of *Pyricularia oryzae* spores, resistance to biotic stresses	[68]
10	*AtDRN1*	*At2g45180*	–	–	Leaf	Response to avirulent bacterial phytopathogen *Pst DC3000*	[69]
11	*AtAZI1*	*At4g12470*	–	–	Root, Leaf	Salt stress tolerance, regulate systemic acquired resistance	[29,38]
12	*OsLTP1*	*CAX20937*	–	–	Leaf, Root, Lemma, Palea, Anther	Structural barriers and organ protection against mechanical disruption and pathogen attack	[70,71]
13	*ZmLTP3*	*Zm00001d043049*	*ZmLTP1.1*	–	Root, Coleoptile, Leaf, Silk, Ovary	Improve plant survival under salt and drought stresses	[37]

Note: The orthologous genes are highlighted with gray background in the table.

To date, at least 15 LTP transporters have been reported to be involved in anther and/or pollen development in *Arabidopsis*, rice, and maize (Table 2 and Figure 4A). The type III LTPs *AtLTPc1* (*At5g07230*), and *AtLTPc3* (*At5g62080*) are required for the exine precursors transport from the tapetal ER to the microspore surface, and also act as the part of components of pollen exine [14]. The mutants of *atltpg3* and *atltpg4* showed deformed or collapsed pollen grains, indicating the essential roles of AtLTPg3 and AtLTPg4 in pollen development [17]. *Zmms44*, the ortholog of *AtLTPc1* and *AtLTPc3*, is the only cloned dominant male sterility gene in maize, functions as a signal peptide, and facilitates the secretion of protein from tapetal cells into the locule [32]. *OsC4*, an ortholog of *AtLTPc1* and *AtLTPc3* in rice, along with *OsC6* are related to tapetal PCD of anther [16,39,40], and OsC6 is reported to be localized in anther extracellular space [16]. In tapetal cytoplasm, OsC6-loaded pollen wall constituents (PWCs) are recruited by OsLTP47, which is plasma membrane-localized, and PWCs were transferred to OsLTP47; then, OsLTP47 recruits OsC6 in anther locules to the tapetal cell membrane and transfers PWCs to OsC6 for further transports into pollen exine [31]. Further, OsC6 is also involved in anther cuticle formation by translocation of cutin precursors from tapetum to the anther outer wall [16]. These data indicate that the orthologous LTP genes play similar roles in different plants, and LTPs may play relay roles in lipid transport during anther and pollen development.

Another example, OsLTPL94/OsEPAD1, determines pollen exine patterning as a microspore membrane recruiting protein, and OsLTPL94/OsEPAD1 is not only secreted by pollen mother cells (PMCs) but also derived from the tapetum [5,15]. ZmLTPg11 and ZmLTPx2, specifically expressed in the microsporocytes, transported the same substrates with their orthologs of OsLTPL94/OsEPAD1 and TaMs1 in rice and wheat; additionally, the pollen development is not effective in the *zmltpg11/zmltpx2* double mutant [33]. These findings suggest that some of the orthologous LTPs, such as OsLTPL94/OsEPAD1, TaMs1, ZmLTPg11, and ZmLTPx2, may play conserved roles in lipid transport essential for pollen wall (exine) development.

Moreover, the function of LTP genes in male fertility might be influenced by abiotic stress. For example, OsDIL/OsLTP6 is related to drought stress tolerance, and OsDIL-overexpressing transgenic plants showed fewer defective anther sacs and less severe tapetal defects when treated with drought at the reproductive stage in rice [41,42]. *AtLTP12* determines site proline biosynthesis, which can restore the fertility of proline-deficient microspores [43]. However, transgenic plants with gain of function for *AtLTP5*, the homolog of *AtLTP12*, shows abnormal vegetative and pollen tube growth, and *AtLTP5* is mainly expressed in root, shoots, leaves, and pistil of *Arabidopsis* [44,45]. MZm3-3 (ZmLTPc2) plays an important role in pollen coat formation, but the mechanism needs to be further studied [34]. Together, LTP transporters may play conserved and divergent roles in lipid transport for anther and pollen development and male fertility in multiple plants.

Transcriptional regulatory pathways that control male fertility through lipid metabolism have been well studied and reviewed in *Arabidopsis*, rice, and maize [47,58]. Many LTP genes are regulated by transcription factors (TFs) essential to male fertility in plants (Figure 4A). Previous studies suggest that *OsGAMYB*, an important component of gibberellin (GA) signaling, regulates early anther development [72]. OsGAMYB directly regulates the expression of LTP genes, such as *OsC6* and *OsC4* [73]. *OsTDR* and *OsPTC1* act downstream of *OsGAMYB*, and OsPTC1 associates with OsTDR to regulate expression of their downstream genes [74]. For example, *OsC6* and *OsC4* work at the downstream of *OsPTC1*, and *OsTDR1* directly regulates the expression of *OsC6* [75,76]. *AtAMS*, the orthologs of *OsTDR* in *Arabidopsis*, directly regulates *At5g62080* (*AtLTPc3*), *At1g66850* (*AtLTP2.5*), and *At3g51590* (*AtLTP1.7*) [77]. OsEAT1 interacts with the OsTDR protein, and directly regulates the expression of *OsLTPL94*, which plays a key role in tapetum and microspore development [15,78]. Taken together, TFs may transcriptionally regulate the expression of LTP genes involved in male fertility directly or indirectly as well as synergistically.

#### 2.4.2. Vegetative and Female Organ Development

Another important function of LTP transporters is related to vegetative and seed development (Table 2 and Figure 4B, C). Several LTP transporters are reported to be involved in vegetative organ development (Figure 4B). For example, *AtLTP1*, a regulator of ethylene response and signaling, takes part in the export of lipids to the plant surface [21,25,59]. *AtLTP2*, as the homolog of *AtLTP1*, plays a major structural role to maintain the adhesion integrating the mainly hydrophobic cuticle and the hydrophilic underlying cell wall [60]. *AtLTP1* to *AtLTP6* play multifunction roles in plant growth and reproduction [45]. In *atltpg2* and *atltpg1 atltpg2* double mutants, the composition of cuticular wax in the stems and siliques was significantly altered [23]. *OsPSD1/OsPTD1*, the ortholog of *AtLTP6* in rice, regulates cell development, plant height, and sensitivity to temperature conditions [22,24].

Some homologous or orthologous LTPs may play similar roles in plant vegetative development between monocots and dicots. AtLTPG23 and AtLTPG26 are related to suberin biosynthesis [3]. AtLSR1 regulates leaf senescence [63].

Additionally, some LTP genes are involved in seed development or seed germination (Figure 4C). For example, OsLTPL36 is essential for seed quality, seed development, and germination in rice [20]. As the ortholog of *OsLTPL36*, *ZmBETL-9* is transcribed in the outer surface of the developing endosperm [35]. *AtEND1* is widely expressed in root, leaf, stem, flower, and seed, and plays important roles in gametophytic tissues and developing endosperm in *Arabidopsis* [62]. AtLTPG5 and AtLTPG6 are involved in cuticle development and seed coat suberization [17]. In summary, LTP transporters play indispensable roles in lipid secretion and transportation in different organs in plants, and thus are critical for plant growth and development.

#### 2.4.3. Biotic and Abiotic Stress Response

Stress resistance is an important characteristic for crop breeding. However, the mechanism of LTPs in the induction of stress resistance is not fully understood [10]. According to the stress factors and functional models, we classified all the reported LTP transporters-mediated stress response into two groups: abiotic stress and biotic stress (Table 2 and Figure 4D).

Abiotic stress factors include hormones, cold, and salt, etc. For example, enhancing abscisic acid (ABA), salicylic acid, and 16-hydroxypalmitic acid could induce the expression of *OsLTP5* in rice [26]. *Zm-LTP*, the ortholog of *OsLTP5* in maize, binds to calmodulin (CaM) in a Ca^2+^-independent manner to modulate its lipid-binding ability [36]. The mutant of *atazi1* is hypersensitive to salt stress, while *AtAZI1*-overexpressing plants are markedly more tolerant [29,38]. OsLTPL159 is involved in cold tolerance at the early seedling stage in rice [30]. ZmLTP3 enhances plant salt tolerance and drought resistance [37]. For signal transport, AtDIR1 and AtDIR1-like are required for the transmission of a mobile signal during systemic acquired resistance [65,79]. For biotic stress, the AtLTP3 contributes to disease susceptibility in *Arabidopsis* by enhancing ABA biosynthesis and the *atltp3* mutant seeds showed impaired germination under salt and osmotic treatments [19,66,67]. In addition, the double mutant of *atltp3*/*atltp4* showed that the susceptibility to *Pseudomonas* is reduced and the ABA biosynthesis genes are down-regulated [45,66]. AtDRN1 is involved in response to the avirulent bacterial phytopathogen *Pst DC3000* [69]. AtLTPG1 could alter the composition of cuticular lipids, increase plastoglobules, and enhance susceptibility to infection by the fungal pathogen *Alternaria brassicicola* [27]. OsLTP1, the first identified LTP gene of rice, plays important roles in structural barriers and organ protection against mechanical disruption and pathogen attack [70,71]. OsLTP110 could inhibit the germination of *Pyricularia oryzae* spores in vitro, and overexpression of *OsLTP110* could bring a substantial resistance to biotic stresses [68]. In summary, some LTP transporters play critical roles during plant response to abiotic and biotic stresses, which will be helpful to improve crop stress resistance by manipulating the related LTP genes in the future.

### 2.5. Functional Prediction of LTP Genes Based on Bioinformatics Analysis

As the biological functions of genes are related to their spatiotemporal expression patterns, bioinformatics analyses will provide useful information for exploring the function of maize LTP genes. They are a useful way to predict unknown gene function by using transcriptomic analyses based on RNA-seq data [47,58,80,81]. In order to predict the functions of maize LTP genes involved in anther and pollen development, we used the RNA-seq data of W23 (from stages S2 to S12), B73, Oh43, and Zheng58 (from stages S5 to S12) developing anthers and qPCR analysis of maize anther from stage S5 to S13 to analyze the expression patterns of all maize LTP genes during anther development (Figure 5).

Based on the RNA-seq data, all the maize LTP genes can be classified into two clusters (I and II). The cluster I contains two subclusters (I-1 and I-2). The subcluster I-1 includes eight LTP genes, which are expressed at early development stages (from stage S2) of maize anthers. Among them, *ZmLTPg11* and *ZmLTPg4* are orthologous to male-sterile genes *OsLTPg29* and *AtLTPG4* [5,15,17], suggesting these two genes may be required for anther development. The subcluster I-2 contains 11 LTP genes, with the expression peaks at middle development stages (from S6 to S10) of maize anthers. There are five anther or tassel specific expression genes (*ZmLTPc1/Zmms44*, *ZmLTPc2/Zmanthr3*, *ZmLTP2.9*, *ZmLTP1.5*, and *ZmLTPd9*) in subcluster I-2 based on the information in MaizeGDB (www.maizegdb.org), and two of them (*ZmLTPc1/Zmms44* and *ZmLTPc2/Zmanthr3*) have been reported to be essential for pollen development. It is reasonable to predict the other three LTP genes may also be required for anther and pollen development, which should be proved with experimental data in the future.

The cluster II contains four subclusters (II-1 to II-4). Subcluster II-1 includes five genes with high expression at late development stages (from S10 to S13) of maize anthers. Among them, *ZmLTPg20* is anther-specific expression gene, and *ZmLTPg15* is orthologous to the male-sterile gene AtLTPG3 [17]. The subcluster II-2 and II-3 contain five LTP genes respectively, which are expressed lower than cluster I or subcluster II-1 in developing anther. Among them, *ZmLTPg26* and *ZmLTPg14* are orthologs of male-sterile genes *OsLTP47* and *AtLTPG4* [17,31], respectively. Subcluster II-4 includes 29 genes with low expression in W23 anthers, whereas some of them show expression peaks at certain anther stages in different inbred lines, such as *ZmLTP2.5* with high expression at stages S7–S9 in B73, Oh43, and Zheng58 anthers, but undetectable expression in W23 anther. This finding indicates that some gene expression might be related to their genetic backgrounds. Notably, five LTP genes (*ZmLTPx2*, *ZmLTPx1*, *ZmLTPd16*, *ZmLTPd15*, and *ZmLTP1.8-1*) are orthologous to the male-sterile genes *OsLTPg29*, *OsC6*, *OsLTP6*, and *AtLTP5/AtLTP12* [5,15,16,41,42,43,44,45], respectively, and *ZmLTPx1* is also a tassel-specific expression gene in maize, implying that these LTP genes may be involved in anther development and male fertility. Furthermore, the expression patterns of anther- or tassel-specific expression genes and the orthologs of GMS genes in *Arabidopsis* and rice are confirmed by qRT-PCR analysis (Figure 5B). These findings indicate that these LTP genes may be required for maize anther development and male fertility, which need to be proved in the future.

### 2.6. Substrate Identification Strategies of Plant LTP Transporters

As previously reported, lipid metabolism plays important roles in plant reproductive development, including anther cuticle and pollen wall development [82], and the LTP transporters play roles in various physiological processes in plants. However, the exact substrates of most LTP transporters are still unclear. Here, we summarize several approaches to identify the substrates of LTP transporters (Table 3), which will be helpful for exploring the molecular mechanism of maize LTP transporters in the future.

The most effective approach to identify the substrates of LTP transporters is a transport assay by PIP lipid strips and membrane lipid strips or isotope labelling, etc. However, those transport assays must be based on the overexpression of the LTP protein in the proper expression systems, such as the prokaryotic expression system. TaMs1 and its homologous genes OsLTPg29/OsLTPL94/OsEPAD1 and ZmLTPg11 in rice and maize, respectively, were confirmed to transport phospholipids in anthers by protein-lipid overlay assay by PIP lipid strips and membrane lipid strips [33,83]. AtDIR1 and AtAZI1 transport azelaic acid and the phosphorylated sugar derivative glycerol-3-phosphate by using isotope labelling of 14C-containing products measured by the TLC method [38]. Moreover, the substrates of some LTP transporters, such as OpsLTP1 and AtLTP3, have been determined by using the quantification of total lipids by spectrophotometric methods [67], and of Lc-LTP2 transporters FAs (C12-C22) and lysolipids by molecular modeling, 2-p-toluidinonaphthalene-6-sulphonate (TNS) displacement, and liposome leakage experiments [84].

The GC-MS system is a convenient and approach to study the potential substrates of LTP transporters by using mutant and wild-type plant tissues, such as anthers, roots and leaves. The substrates of at least 12 LTP transporters have been predicted by using this approach in *Arabidopsis*, rice, and maize (Table 3). One advantage of this approach is that mixtures such as the content of cytosol in plants can be directly employed, and subsequently the isolated compounds can be used as a direct proof. Thus, this is a very powerful approach to identify the substrate of LTP transporters, although the conclusion is not very convincing. For example, OsLTP47 is reported to be essential for transporting the anther sporopollenin precursors, and fatty acid content was reduced in *osltp47* mutant lines compared to WT due to the lipidic analysis of the wild-type and mutant mature anthers by using GC-MS [31]. In a word, the substrate identification of plant LTP transporters has made great progress based on different strategies, while the translocated substrates and detailed transport mechanism of the majority of LTP proteins remain unexplored, which need to be investigated in the future.

## 3. Discussion

LTP transporters play pivotal roles in multiple biological processes, such as abiotic and biotic stress response, plant signal transduction, and biosynthesis of lipid polymer sporopollenin and protective water-impermeable barriers in different plant organs [1,12,17,28,85]. Compared with the relatively comprehensive and systematic in-depth studies of LTP genes in *Arabidopsis* and rice, the functional mechanisms of maize LTP genes remain largely unknown. Here, we presented a preliminary introduction and characterization of the whole 65 LTP genes in maize. There are at least four significant points compared with previous studies [13]. First, we analyzed the basic characteristics and spatial expression patterns of all the 65 ZmLTP genes, which were identified based on the up-to-date maize B73 reference genome information, and seven genes were different from the previous study. Moreover, we performed a phylogenetic and gene ontology analysis of the LTP transporters in maize, which provides a basic framework for future research on maize LTP genes. Second, we summarized the research progress of the LTP transporters involved in diverse biological processes in model plants *Arabidopsis* and rice, such as anther and pollen development, vegetative organ development, seed development and germination, and biotic and abiotic stress response, which provides a mode to study the unknown LTP transporters in maize based on the functional conservation of LTP orthologs during plant evolution, together with gene ontology analysis. Third, we predicted the potential functions of maize LTP genes involved in anther development by using transcriptomic analysis based on RNA-seq and qRT-PCR assays. These findings provide useful clues for functional investigation of LTP transporters in maize. Finally, we summarized the advances and challenges in substrate identification of plant LTP transporters, and presented the future research directions and potential applications of LTP proteins in crop molecular breeding.

## 4. Materials and Methods

### 4.1. Identification of LTP Genes in Maize

Gene IDs of the LTP family in *Arabidopsis*, rice, and maize were acquired in previous studies [4,13]. The amino acid sequences of *Arabidopsis*, rice, and maize LTPs were accepted on NCBI (https://www.ncbi.nlm.nih.gov/, assessed on 30 October 2022), EnsemblPlants (http://plants.ensembl.org/index.html, assessed on 30 October 2022), and MaizeGDB (https://maizegdb.org/, assessed on 30 October 2022), respectively. Physical locations including B73 RefGen_v3, Zm-B73-REFERENCE-GRAMENE-4.0, and Zm-B73-REFERENCE-NAM-5.0, and expression patterns for B73 RefGen_v3 of maize LTP genes were acquired on MaizeGDB (https://maizegdb.org/, assessed on 25 October 2022). To collect information of physical and chemical properties for maize LTP proteins, Expasy (https://web.expasy.org/protparam/, accessed on 25 October 2022) was used to determine molecular weight and theoretical pI, and Cell-PLoc-2.0 (http://www.csbio.sjtu.edu.cn/bioinf/Cell-PLoc-2/, accessed on 25 October 2022) was used to predict subcellular localization.

### 4.2. Phylogenetic and GO Analysis of Maize LTP Genes

Multiple alignments of maize, rice, and *Arabidopsis* LTP amino acid sequences were performed by ClustalW of MEGA7. Phylogenetic trees were constructed with MEGA7 using the maximum likelihood method, and bootstrap values were based on 500 replicates. The GO analysis was performed by GENE ONTOLOGY (http://geneontology.org/, accessed on 2 November 2022).

### 4.3. RNA Extraction, qPCR, and RNA-Seq Analyses

Total RNA of B73 anthers at stages S5 to S13 were isolated by using a Trizol reagent (Invitrogen, Waltham, MA, USA). The reverse transcription (RT) reaction was operated according to the protocol of the RT system (TransGen, Beijing, China).

RNA-sequencing (RNA-seq) analysis of maize anthers from stages S2 to S13 for W23, and S5 to S13 for B73, oh43, and Zheng58 were carried out as described by Jiang et al. [81].

For quantitative real time-PCR (qPCR), three technical replicates and three biological replicates were performed on each sample, and data were analyzed by using the 2^-DDCt^ method; each data point is the mean from three replicates ± SD. *ZmUbi2* (*Zm00001d05383*) were used as the internal controls. The primers of qPCR analysis are shown in Appendix A.

## 5. Conclusions

In conclusion, the findings presented here will shed light on our understanding of the critical roles of maize LTPs involved in various biological processes, including the transport and localization of lipidic precursors for anther development and male fertility, vegetative organ and seed development, and signal transport for biotic and abiotic stress response and resistance. Notably, the predicted biological functions of maize LTPs can be verified by using reverse genetics such as CRISPR/Cas9 or RNAi mutagenesis analysis [81,86,87]. With the advances of LTP gene cloning and functional characterization, this study will be an excellent gene resource for improvement of the grain yield, seed quality, and stress tolerance in maize via manipulating the related LTPs, such as through the multiple-control sterility system [54,88], dominant male sterility system [89,90], and other genetic strategies [91,92]. Therefore, this study provides a basic framework for functional research and the potential application of LTPs in multiple plants, especially for male sterility research and application in maize.

## Figures and Tables

**Figure 1 ijms-24-01660-f001:**
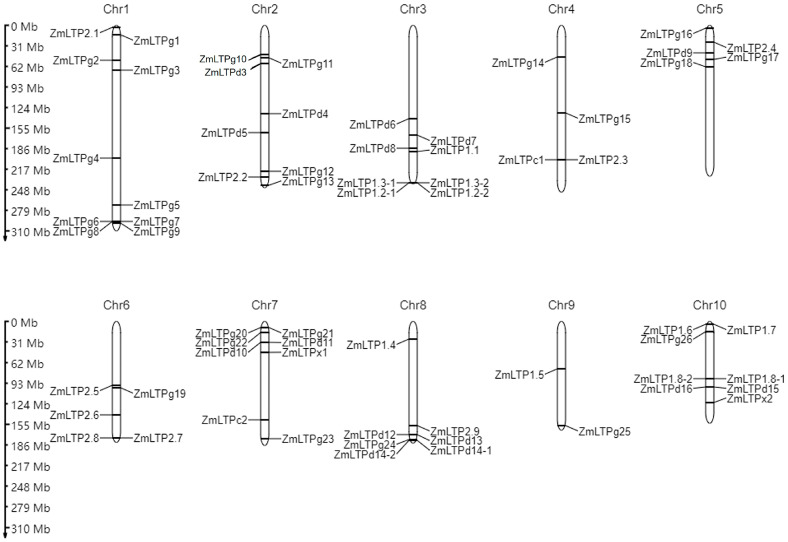
Genomic distribution of the 65 *ZmLTP* genes in maize chromosomes. Chromosome numbers are indicated at the top of each chromosome, respectively. The physical location of each *ZmLTP* gene is indicated to the right of chromosomes. Mb, million base pair.

**Figure 2 ijms-24-01660-f002:**
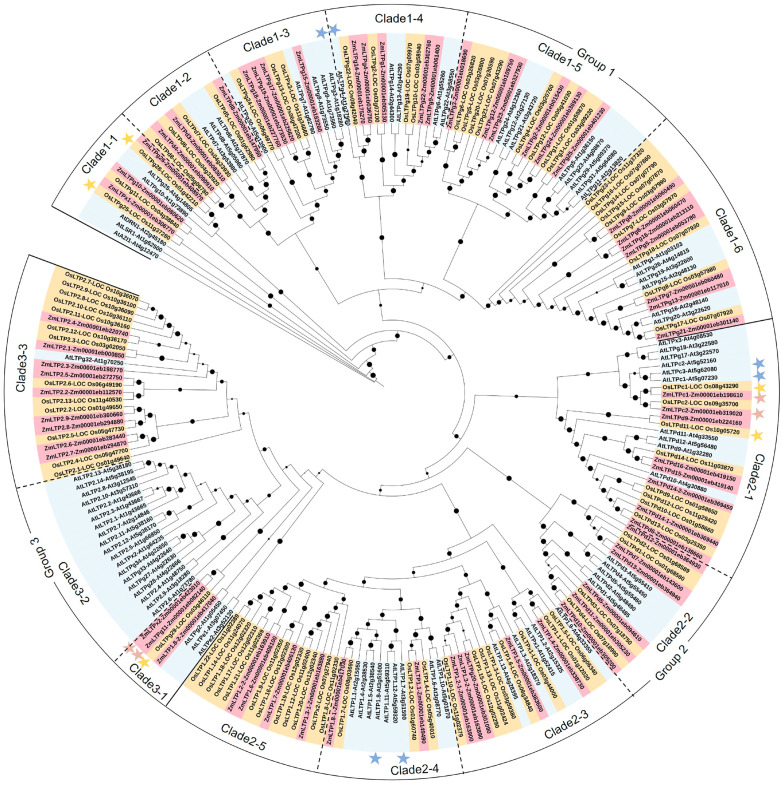
Phylogenetic analysis of LTPs in maize (65), rice (80), and *Arabidopsis* (84). Group 1 can be further classified into six clades (Clades 1-1 to 1-6), which contains 91 LTPs, including 28 in maize, 31 in rice and 32 in *Arabidopsis*. Group 2 can be further classified into five clades (Clades 2-1 to 2-5), which contain 89 LTPs, including 25 in maize, 35 in rice, and 29 in *Arabidopsis*. Group 3 can be further classified into three clades (Clades 3-1 to 3-3), and contains 91 LTPs, including 28 in maize, 31 in rice, and 32 in *Arabidopsis*. Gray, yellow, and violet background represent the LTPs in *Arabidopsis*, rice, and maize; and gray, yellow, and violet asterisks represent the male fertile genes in *Arabidopsis*, rice, and maize, respectively. The dark spots on the branches of the tree indicate the evolutionary distance; the larger dark spot means the closer evolutionary distance of LTPs.

**Figure 3 ijms-24-01660-f003:**
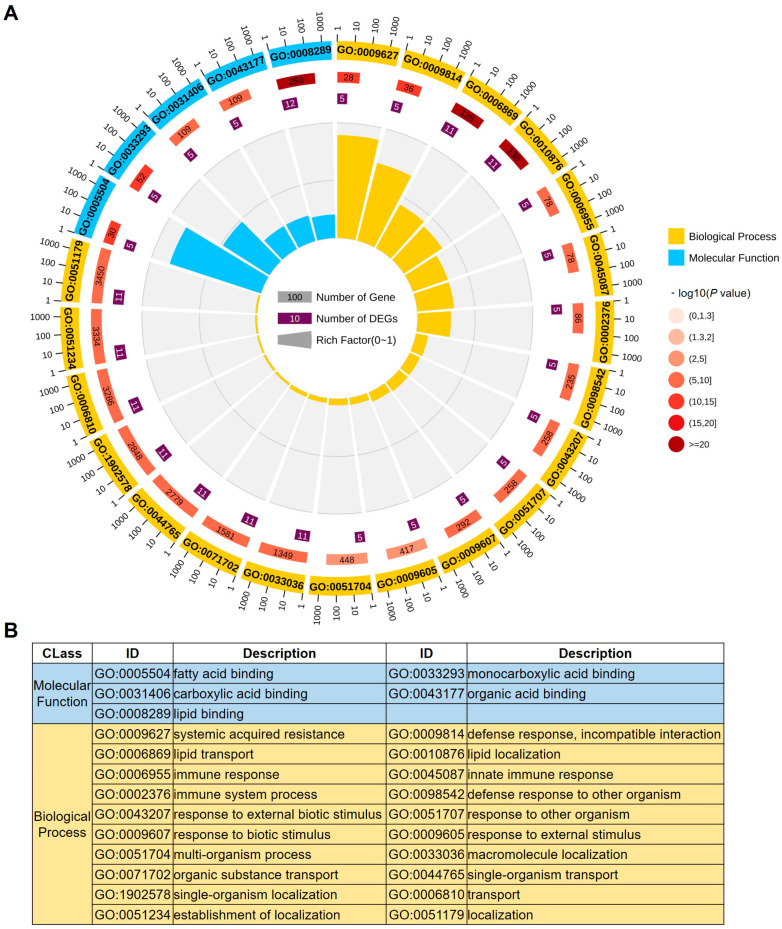
GO analysis of *LTP* genes in maize. (**A**) GO analysis, including rich factor values of each GO term, the number and P value of this GO term in background genes, and GO term of enrichment. (**B**) The ID and description of each GO term.

**Figure 4 ijms-24-01660-f004:**
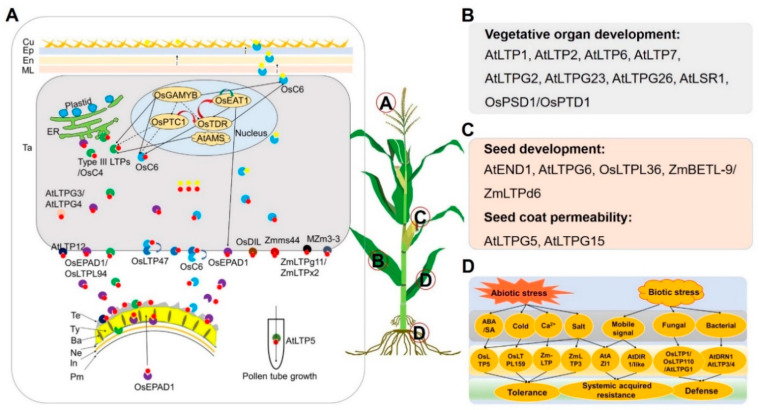
The main functions of LTP transporters in *Arabidopsis*, rice, and maize. (**A**) The proposed working model of LTP transporters involved in anther and pollen development. Several LTPs transfer tapetum-generated lipidic and other precursors for anther cuticle and pollen wall development, and AtLTP5 is required for the apical accumulation of reactive oxygen species in growing pollen tubes. Ba, baculum; Cu, cuticle; Ep, epidermis; En, endothecium; In, intine; ML, middle layer; Ne, nexine; Pm, plasma membrane; Ta, tapetum; Te, tectum; Ty, tryphine. (**B**,**C**) LTP genes participate in vegetative organ or seed development and seed coat permeability in *Arabidopsis*, rice, and maize. (**D**) Regulation of plant resistance by LTP transporters in response to biotic and abiotic factors. The stress factors, LTP transporters, and their corresponding substrates and physiological roles in plants are displayed in (**D**). ABA, abscisic acid; SA, salicylic acid. A–D in red circle represent the mainly expression position of the genes in A to D, respectively.

**Figure 5 ijms-24-01660-f005:**
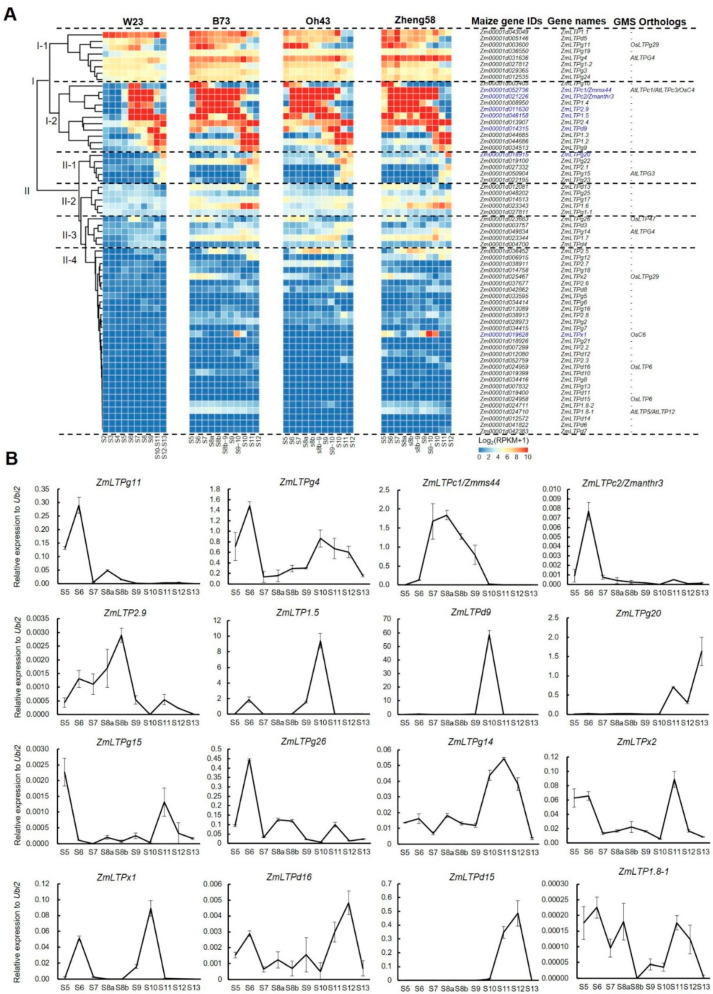
Expression analysis of LTP genes in maize. (**A**) Expression analysis of LTP genes in maize based on RNA-seq data in maize inbred lines W23, B73, Oh43, and Zheng58. These genes are clustered into two clusters. Genes in cluster I and II were clustered into two and four sub-clusters, respectively. The blue font indicates anther- or tassel-specific expression genes. (**B**) qPCR analysis of 16 LTP genes, which are tassel-specific expression, GMS or orthologs of rice and *Arabidopsis* GMS genes from stages 5 to 13 (S5–S13). Data are means SD, *n* = 3.

**Table 1 ijms-24-01660-t001:** The maize LTP family: *LTP* genes based on B73 reference genome V5.0 and expression patterns.

No.	Name	Gene ID (B73 V5) ^1^	Expression Patterns ^2^											
			Root	Leaf	Sem and sam	Internode	Tassel	Anther	Silk	Cob	Seed	Endosperm	Enbryo	Pericarp
1	*ZmLTP1.1*	Zm00001eb149490	−	+	+	−	+	−	−	+	+	−	+	−
2	*ZmLTP1.2-1*	Zm00001eb163900	−	+	−	+	+	−	−	+	+	−	+	−
3	*ZmLTP1.2-2*	Zm00001eb163910												
4	*ZmLTP1.3-1*	Zm00001eb163880	−	+	−	+	+	−	+	−	+	−	−	−
5	*ZmLTP1.3-2*	Zm00001eb163890												
6	*ZmLTP1.4*	Zm00001eb338730	−	+	−	−	+	−	−	−	−	−	−	−
7	*ZmLTP1.5*	Zm00001eb383600	−	−	−	−	+	−	−	−	−	−	−	−
8	*ZmLTP1.6*	Zm00001eb406100	−	+	−	−	−	−	+	−	+	−	+	−
9	*ZmLTP1.7*	Zm00001eb406130	−	−	+	−	+	−	−	+	+	+	+	−
10	*ZmLTP1.8-1*	Zm00001eb417030	−	−	−	−	−	−	−	−	+	+	−	−
11	*ZmLTP1.8-2*	Zm00001eb417040												
12	*ZmLTP2.1*	Zm00001eb000850	−	+	−	−	−	+	−	−	+	−	−	+
13	*ZmLTP2.2*	Zm00001eb112570	−	−	−	−	−	−	−	−	+	+	−	+
14	*ZmLTP2.3*	Zm00001eb198770	−	−	−	−	−	−	−	−	+	+	−	+
15	*ZmLTP2.4*	Zm00001eb220740	−	+	−	−	−	+	−	−	+	−	+	−
16	*ZmLTP2.5*	Zm00001eb272750	−	+	−	−	−	−	−	−	−	−	+	−
17	*ZmLTP2.6*	Zm00001eb283440	+	−	+	−	−	−	−	−	+	−	−	−
18	*ZmLTP2.7*	Zm00001eb294870	+	+	−	+	−	−	−	−	−	−	−	−
19	*ZmLTP2.8*	Zm00001eb294880	+	+	−	+	+	−	−	−	+	−	−	−
20	*ZmLTP2.9*	Zm00001eb360660	−	−	−	−	+	−	−	−	−	−	−	−
21	*ZmLTPc1*	Zm00001eb198610	−	−	−	−	+	−	−	−	−	−	−	−
22	*ZmLTPc2*	Zm00001eb319020	−	−	−	−	+	−	−	−	−	−	−	−
23	*ZmLTPd3*	Zm00001eb083610	−	+	−	−	−	−	−	−	+	−	−	−
24	*ZmLTPd4*	Zm00001eb090910	+	+	−	+	+	−	−	−	−	−	−	−
25	*ZmLTPd5*	Zm00001eb094610	+	+	−	+	+	−	−	−	+	−	+	+
26	*ZmLTPd6*	Zm00001eb138660	−	−	−	−	−	−	−	−	+	+	+	+
27	*ZmLTPd7*	Zm00001eb143600	+	−	−	−	−	−	−	−	−	−	−	−
28	*ZmLTPd8*	Zm00001eb147950	+	+	−	+	+	−	−	−	+	−	−	−
29	*ZmLTPd9*	Zm00001eb224160	−	−	−	−	+	−	−	−	−	−	−	−
30	*ZmLTPd10*	Zm00001eb305200	+	+	−	+	−	−	−	−	−	−	−	−
31	*ZmLTPd11*	Zm00001eb305230	+	+	−	+	−	−	−	−	−	−	−	−
32	*ZmLTPd12*	Zm00001eb364930	+	−	−	+	−	−	−	−	−	−	−	−
33	*ZmLTPd13*	Zm00001eb364940	+	+	−	+	+	−	−	+	+	+	+	+
34	*ZmLTPd14-1*	Zm00001eb369440	−	−	−	−	−	−	−	−	+	+	+	+
35	*ZmLTPd14-2*	Zm00001eb369450												
36	*ZmLTPd15*	Zm00001eb419140	−	−	−	−	−	−	−	−	+	+	−	+
37	*ZmLTPd16*	Zm00001eb419150	−	−	−	−	−	−	−	−	+	+	−	+
38	*ZmLTPg1-1*	Zm00001eb005130	−	+	−	+	+	−	+	−	+	+	+	+
39	*ZmLTPg2*	Zm00001eb015430	−	+	−	−	−	−	+	+	+	−	−	+
40	*ZmLTPg3*	Zm00001eb018690	+	+	−	+	−	−	+	+	+	+	+	+
41	*ZmLTPg4*	Zm00001eb036750	−	+	−	−	+	−	+	+	+	+	+	+
42	*ZmLTPg5*	Zm00001eb053780	+	+	−	−	−	−	−	−	−	−	−	−
43	*ZmLTPg6*	Zm00001eb060470	+	+	−	−	+	+	−	−	−	−	−	−
44	*ZmLTPg7*	Zm00001eb060480	+	+	−	−	+	−	−	−	+	−	−	−
45	*ZmLTPg8*	Zm00001eb060490	+	+	−	−	+	−	−	−	−	−	−	−
46	*ZmLTPg9*	Zm00001eb061400	−	+	−	−	+	+	−	−	−	−	−	+
47	*ZmLTPg10*	Zm00001eb080620	−	+	−	−	+	+	−	−	+	−	−	−
48	*ZmLTPg11*	Zm00001eb082140	−	−	−	−	+	−	−	−	+	+	−	−
49	*ZmLTPg12*	Zm00001eb109750	+	−	−	−	−	−	−	−	−	−	−	−
50	*ZmLTPg13*	Zm00001eb117010	+	+	−	−	+	−	−	−	−	−	−	−
51	*ZmLTPg14*	Zm00001eb175270	−	−	−	−	+	−	−	−	−	+	+	+
52	*ZmLTPg15*	Zm00001eb183260	−	+	−	−	+	−	−	−	−	−	+	−
53	*ZmLTPg16*	Zm00001eb213110	+	+	−	−	+	−	−	−	−	−	−	−
54	*ZmLTPg17*	Zm00001eb225620	+	+	+	+	+	+	−	+	+	+	+	+
55	*ZmLTPg18*	Zm00001eb227760	+	+	+	+	−	−	+	−	+	−	−	−
56	*ZmLTPg19*	Zm00001eb273530	+	+	+	+	+	+	+	+	+	+	+	+
57	*ZmLTPg20*	Zm00001eb301090	−	−	−	−	−	+	−	−	−	−	−	−
58	*ZmLTPg21*	Zm00001eb301140	+	+	−	−	+	−	−	−	+	−	−	−
59	*ZmLTPg22*	Zm00001eb302760	+	+	+	+	+	−	−	−	+	+	−	+
60	*ZmLTPg23*	Zm00001eb327930	−	+	−	−	−	+	+	−	−	−	−	−
61	*ZmLTPg24*	Zm00001eb369130	−	+	+	−	+	−	+	+	+	−	−	−
62	*ZmLTPg25*	Zm00001eb401230	+	+	+	+	+	−	−	−	+	+	−	+
63	*ZmLTPg26*	Zm00001eb408970	+	−	−	−	+	−	−	−	−	−	+	+
64	*ZmLTPx1*	Zm00001eb306770	−	−	−	−	+	−	−	−	−	−	−	−
65	*ZmLTPx2*	Zm00001eb423010	−	−	−	−	−	−	−	−	−	+	−	−

Notes: ^1^. Gene ID was based on Zm-B73-REFERENCE-NAM-5.0; ^2^. The expression information of maize LTP genes was based on B73 RefGen_v3; ^1^ and ^2^ retrieved from MaizeGDB (www.maizegdb.org, accessed on 10 October 2022). “+” and “−” indicate whether or not the gene was expressed in the corresponding tissue, and “+” was highlighted with gray background.

**Table 3 ijms-24-01660-t003:** The substrate identification of plant LTP transporters.

No.	LTP Transporters	Substrate(s)	Method of Substrate Identification	References
1	TaMs1	Phospholipid	Purification of the fusion proteins of MBP-TaMs1-His in *E. coli* and protein-lipid overlay assay by PIP lipid strips and membrane lipid strips	[33,83]
2	OsLTPg29/OsLTPL94/OsEPAD1	Phospholipid	Purification of the fusion proteins of MBP-OsLTPg29-His in *E. coli* and protein-lipid overlay assay by PIP lipid strips and membrane lipid strips	[33]
3	ZmLTPg11	Phospholipid	Purification of the fusion proteins of MBP-ZmLTPg11-His in *E. coli* and protein-lipid overlay assay by PIP lipid strips and membrane lipid strips	[33]
4	AtDIR1	Acid azelaic acid; phosphorylated sugar derivative glycerol- 3-phosphate	^14^C-containing products measured by TLC method	[38]
5	AtAZI1			
6	OpsLTP1	16C and 18C fatty acids, linoleic acid	Quantification of total lipids by spectrophotometric methods	[67]
7	AtLTP3	16C and 18C fatty acids	Quantification of total lipids by spectrophotometric methods	[67]
8	Lc-LTP2	FAs (C12-C22) and lysolipids	Molecular modeling, 2-p-toluidinonaphthalene-6-sulphonate (TNS) displacement and liposome leakage experiments	[84]
9	AtLTP1	Cuticular wax	Substance analysis by using GC-MS system	[21]
10	AtLTP2	Cuticular wax	Substance analysis by using GC-MS system	[60]
11	AtLTP3	Fatty acid	Substance analysis by using GC-MS system	[19]
12	AtLTPG1	Cuticular wax	Substance analysis by using GC-MS system	[27]
13	AtLTPG2	Cuticular wax	Substance analysis by using GC-MS system	[23]
14	AtLTPG4	Cuticular wax	Substance analysis by using GC-MS system	[17]
15	AtLTPG6	Cuticular wax	Substance analysis by using GC-MS system	[17]
16	OsLTPL36	Fatty acid	Substance analysis by using GC-MS system	[20]
17	OpsLTP1	Fatty acid	Substance analysis by using GC-MS system	[67]
18	OsLTP5	Fatty acid	Substance analysis by using GC-MS system	[26]
19	AtLTPG15	Suberin monomer	Substance analysis by using GC-MS system	[61]
20	OsLTP47	Fatty acid	Substance analysis by using GC-MS system	[31]

## Data Availability

All data are shown in the main manuscript and in the Appendix A.

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
