# Peer review of "A Systematic Investigation of Lipid Transfer Proteins Involved in Male Fertility and Other Biological Processes in Maize"

_ijms, 2023, doi:10.3390/ijms24021660_

Round 1

Reviewer 1 Report

This is a study and review of plant lipid transfer proteins (LTPs). The authors were comprehensive and detailed, highlighted the important functions of LTPs in plant fertility, but also mentioned other functions such as abiotic stress and plant growth and development. The authors used well-studied LTPs function in rice and Arabidopsis to predict the function of orthologous genes in maize.In general, the authors have done a good job, but there are still some details that need to be improved.

1. Table 1, The author does not indicate the meaning of +,-. If it is to indicate whether the gene is expressed in different tissues, it is recommended to directly label the value of FPKM or TPM and indicate the application (if any). And it is easy to distinguish if marked with different background colors.

2. Line 15, most --> largest

3. Figure 2. The meaning of different colors and asterisks should be specified.

4. Line 64, GO (Gene Ontology) should be Gene Ontology (GO).

5. Line 67, Gene Ontology (GO) should be GO.

6. Lines 68 and 82, Numbers don't usually appear directly at the beginning of a sentence. This issue should also be noted elsewhere in the text.

7. Table 2. Does the gray background in the table indicate that they are orthologous genes? Please indicate its meaning after the form.

Reviewer 2 Report

This paper analyzed the evolutionary relationship and function of maize LTP gene family in male fertility and other processes. The author makes a good bio-informatics analysis, and the results are meaningful. However, this paper is like both an article and a review. As an article, the research depth is insufficient. At the same time, the research content of this paper is very similar to that of a previously published paper, especially for the sequence characteristics, tissue expression and GO pathway analysis of LTP gene family members, which leads to the low innovation of this paper. This paper is Wei, K.; Zhong, X., Non-specific lipid transfer proteins in maize. BMC Plant Biol 2014, 14, (1), 281, Doi: 10.1186 / s12870-014-0281-8. Considering this paper's in-depth bio-informatics analysis and discussion of the functional research and potential application of LTPs in maize, I suggest the revised paper can be published on IJMS with a major revise.

1. I think this article belongs in review. If the author insists on submitting an article, the structure of the paper should be carefully adjusted, the experimental part should be displayed in the results, and some contents should be discussed, especially the part 2.4, which does not have any data analysis and is just a review of the research progress, and should not be included in the results. Similarly, there are other contents. Please clarify the results of this article carefully, and do not confuse it with review.

2. Since the author focused on the role of LTPs in male fertility of maize, the introduction should focus on the research progress of the function of LTP genes involved in male fertility of maize, Arabidopsis and rice. In my opinion, the introduction in the literature review is insufficient.

3. In the part of 2.1. The Nomenclature and Characterization of Maize LTP genes, The author's analytical work is not new compared with that of the predecessors. Based on the newly annotated maize genome, the authors identified 65 LTPs genes, which is two more than in previous studies. I suggest that the authors should focus on introducing the characters and functions of these two additional LTPs genes to the readers, so that we can have an in-depth understanding of the new members of maize LTPs.

4. In the section on evolutionary tree construction, Materials and Methods refer to the proximity method, but the Results section refers to the maximum likelihood method. What is the method used to construct the evolutionary tree?

5. The dark spots on the branches of the tree are unexplained

6. In the validation section of fluorescent quantitative PCR, how the authors selected LTPs for quantitative analysis. Based on the analysis of transcriptome data, the applicant seems to have missed some fluorescence quantitative analysis of differentially expressed LTPs. Please add.

7. Furthermore, the expression patterns of anther- or tassel-specific expression genes, and the orthologs of GMS genes in Arabidopsis and rice are in good agreement with the qRTPCR results (Figure 5B).

8. Please add some explanation for this part of the results, I have not seen some data, such as the supplement figure or table

9. In the discussion section, the authors were asked to explain the differences and novelty in the analysis of LTPs gene family members, metabolic pathways, and transcriptional regulation from previous similar studie.

10. There are some mistakes in writing and grammar, such as Ca2+. Due to the lack of line numbers in the paper, it is impossible to point out one by one, the full text needs to be carefully revised

Round 2

Reviewer 2 Report

the author revised the manuscript according to the review's comments, I think it can be accepted now